# Two Patients with Spontaneous Spinal Epidural Hematoma Carrying a Good Prognosis without Surgical Operations

**Katsuhiko Ogawa** [1,2,*], **Takayoshi Akimoto** [1] , **Makoto Hara** [1], **Midori Fujishiro** [3], **Hiroshi Uei** [4]
and **Hideto Nakajima** [1]

1  Division of Neurology, Department of Medicine, Nihon University School of Medicine, Tokyo 173-8610, Japan
2  Department of Neurology, Akabane Central General Hospital, Tokyo 115-0044, Japan
3  Division of Diabetes and Metabolic Diseases, Department of Medicine, Nihon University School of Medicine, Tokyo 173-8610, Japan
4  Department of Orthopaedic Surgery, Nihon University School of Medicine, Tokyo 173-8610, Japan
*  Correspondence: ogawa.katsuhiko@nihon-u.ac.jp; Tel.: +81-3-3972-8111; Fax: +81-3-3972-3059

**Abstract:** (1) Introduction: Spontaneous spinal epidural hematoma (SSEH) points to hematoma within the epidural space of the spinal cord without traumatic or iatrogenic causes. (2) Case Reports: One patient showed paraplegia, numbness of both legs with acute onset, acute myelopathic signs, subsequent to back pain. Magnetic resonance imaging (MRI) showed hematoma in the posterior part of the thoracic spinal cord. Another patient showed acute numbness in the shoulder, upper part of the back, and the upper extremity on the right side after pain in the back, shoulder, and neck on the right side. Sagittal computed tomography (CT) images of the cervical bone showed a high-density area behind the spinal cord between C4 and C7. MRI analysis showed hematoma in the right diagonally posterior part of the cervical spinal cord. These 2 patients lacked traumatic or iatrogenic events, and their symptoms abated without surgical operation. (3) Conclusions: The location of hematoma correlated with symptoms in each patient. SSEH is rare but should be taken into account in patients with myelopathy or radiculopathy with acute onset subsequent to back pain. The usefulness of emergent CT scans of the spinal cord prior to MRI analysis was shown in the diagnosis of SSEH.

**Keywords:** acute onset; back pain; CT; MRI; myelopathy; radiculopathy; spontaneous spinal epidural hematoma

## 1. Introduction

Spontaneous spinal epidural hematoma (SSEH) is a rare disease that affects only 1 in 100,000 people per year [1–9]. SSEH occurs predominantly in patients between the ages of 40 and 80 years, with a mean age of 71 years [10]. Men are slightly more frequently affected than women [1,5,7]. SSEH can produce devastating neurological deficits such as severe paraplegia and sphincter-rectal dysfunction [1], and this condition points to the formation of hematoma caused by the collection of blood within the epidural space of the spinal cord in the absence of traumatic or iatrogenic causes [1–6,10–14]. It is characterized by progressive palsy of the extremities, sensory disturbances, and sphincter-rectal dysfunction. These clinical features are subsequent to severe neck or back pain with acute onset, which are caused by compression of the spinal cord and the nerve radicles [1–6,8,10,12–14]. In principle, Emergent surgical treatment is indicated in patients with progressive neurological deficits [10]. Evacuation of the hematoma for decompression of the spinal cord remains a standard surgical procedure [10]. Early and individualized rehabilitation programs after surgical operation are beneficial for patients with SSEH as well [10]. For the diagnosis of SSEH, immediate investigation using magnetic resonance imaging (MRI) is useful and provides detailed information on the location of a hematoma and compression of the spinal cord [3,8]. Provoking factors associated with the incidence of SSEH include the use of anticoagulant or antiplatelet drugs, minor trauma, peridural catheter insertion,

severe cough, hemophilia, neoplasms, arteriovenous malformation, hypertension, and sneezing [8,9].

Here, 2 patients who were diagnosed as having SSEH are reported (Table 1). One patient showed acute myelopathy. In contrast, the other patient showed acute radiculopathy. Clinical features and radiological findings in these 2 patients are discussed.

**Table 1.** Characteristics of 2 patients with spontaneous spinal epidural hematoma.

| Patient Number | Age/ Sex | Type of Onset | Chief Complaints | Radicular Signs | Myelopathic Signs | Surgical Operation | Prognosis | Vascular Risk Factors | Anti- Thrombotic or Platelet Drug |
|---|---|---|---|---|---|---|---|---|---|
| 1 | 88/f | acute | back pain | - | paraplegia, numbness below bil. L1, decrease of micturition | - | good | HTN | aspirin |
| 2 | 55/f | acute | neck pain, pain in rt shoulder/ back | numbness of rt arm/ shoulder, paralysis of rt fingers | - | - | good | - | - |

Abbreviations: bil, bilateral; f, female; HTN, hypertension; rt, right.

## 2. Case Presentation

### 2.1. Patient 1

An 87-year-old woman felt back pain with acute onset in the morning and then noted numbness and paralysis of both legs and a decrease in micturition in the afternoon of the same day. In the evening, she was admitted to the emergency room of a nearby hospital (Akabane Central General Hospital). Hypertension, cerebral infarction, bronchial asthma, and sleep apnea syndrome had been recorded in her medical history. Aspirin (100 mg/day) and antihypertensive drugs had been administered. On admission, paraplegia of a moderate level, numbness of both lower extremities, and a decrease in micturition were noted. She was diagnosed as having acute myelopathy, based on the neurological findings, and then transferred to the Neurology Ward in the night of the same day (Nihon University Hospital). On admission, she was hypertensive (164/98 mmHg) and afebrile (36.4 °C). General internal medical findings were normal. On neurological examination, she was conscious and showed good orientation regarding names, places, and dates. There were no abnormalities in the cranial nerves including movements of eyes, facial muscle power, facial sensation, hearing, and tongue. Dysarthria was not shown. Muscle strength of the upper extremities was normal; however, moderate paraplegia was noted. Deep tendon reflex was decreased in the upper extremities and was accelerated in both lower extremities. Bilateral Babinski's sign was negative. Incoordination of the upper extremities was not noted. Muscle tone of the four extremities was normal. Objective sensation for pain and soft touch was normal. Subjective numbness below the level of L1 was noted on both sides. Micturition was decreased.

MRI of the spinal cord analysis was performed because the presence of acute lesions on the spinal cord was suspected, based on neurological findings and the clinical course. As a result, sagittal T1-weighted MRI showed an iso-intensity mass in the epidural space behind the spinal cord at the level from Th3 to Th6 (Figure 1A). Compression of the spinal cord was shown at the level between Th4 and Th6 (Figure 1A,B). The lesion was delineated as a high-intensity mass with heterogeneity and a superiorly adhered iso-intensity region in sagittal T1-weighted images (Figure 1B). Axial T1-weighted MRI showed an iso-intensity mass in the posterior part of the epidural space and a deviation of the spinal cord to the anterior direction (Figure 2A). The lesion was delineated as a heterogeneous high-intensity mass in axial T2-weighted MRI images (Figure 2B).

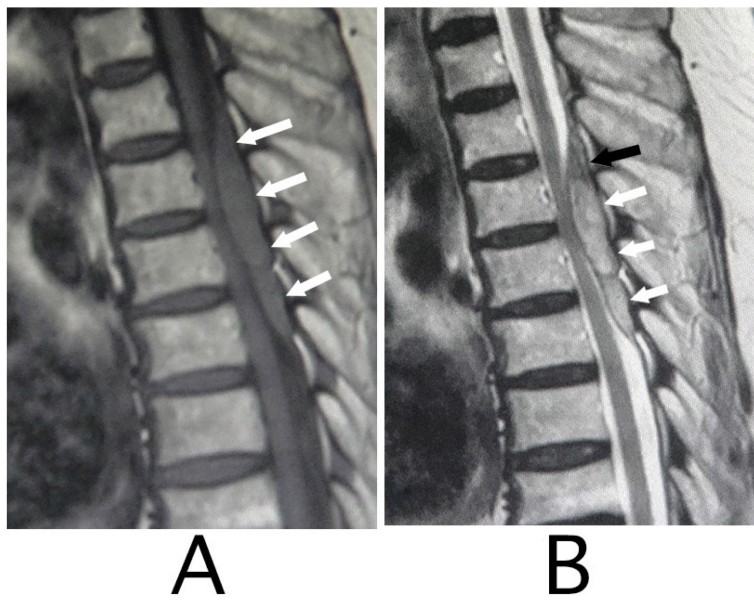

**Figure 1.** Sagittal MRI. An iso-intensity mass in the epidural space was shown behind the spinal cord at the level from Th3 to Th6 (white arrows) (T1-weighted image; (**A**)). The lesion was observed as a heterogeneous high-intensity mass (white arrows) and a superiorly adhered iso-intensity region (T2-weighted image; a black arrow) (**B**).

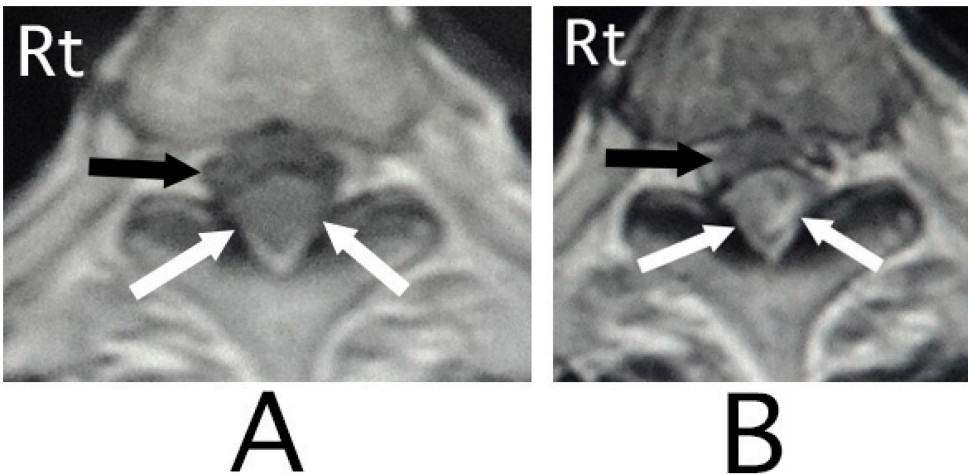

**Figure 2.** Axial MRI. An iso-intensity mass was identified in the posterior part in the epidural space (white arrows) (T1-weighted image; (**A**)). The lesion was delineated as a heterogeneous high-intensity mass (white arrows) (T2-weighted image; (**B**)). The spinal cord was compressed in the anterior direction (black arrows) (**A**,**B**). (Rt = right).

Based on MRI findings and the clinical course, she was diagnosed as having SSEH and hospitalized. Intravenous administration of methylprednisolone (1000 mg/day) was performed for 3 consecutive days, followed by oral administration of prednisolone at 50 mg/day. The amount of prednisolone was decreased by 10 mg/day every 5 days. After hospitalization, paraplegia, numbness, and decreased micturition gradually improved, and then, these neurological symptoms disappeared on day 4 of hospitalization, but mild ataxic gait was recorded. After that, she needed assistance while walking because of ataxic gait. Ataxic gait gradually improved, and then disappeared by day 20 of hospitalization. After discharge, disappearance of the hematoma was confirmed in a follow-up MRI analysis.

*2.2. Patient 2*

A 55-year-old woman complained of pain in the region from the shoulder and the back on the right side to the back of the neck with acute onset in the morning. After that, she felt numbness of the shoulder and upper extremity and palsy of the distal part of the extremity on the right side. She was admitted to the Neurology Ward (Nihon University Hospital) at 12 pm on the same day. There had not been significance in her past history. On admission, she was non-hypertensive (134/72 mmHg) and afebrile (36.2 °C). General internal medical findings were normal. She was conscious and showed good orientation regarding names, places, and dates. There were no abnormalities in the cranial nerves including movements of eyes, facial muscle power, facial sensation, hearing, and tongue. Muscle strength of the four extremities was normal. Deep tendon reflex was normal. Bilateral Babinski's sign was negative. Muscle tone of the four extremities was normal. Incoordination of the four extremities was not shown. Truncal and gait ataxia was also not shown. Micturition was not shown as well. Objective sensation of pain and soft touch was normal; however, subjective numbness of the shoulder, upper part of the back, and the upper extremity on the right side were shown. The numbness was aggravated when she bent the trunk in the anterior direction. Mild palsy of the right fingers was noted.

The lesion in the intracranial region or the cervical spinal cord was suspected. Computed tomography (CT) images of the head and X-ray of the cervical bone were normal; however, sagittal CT images of the cervical bone showed a slightly abnormal high-density area (HDA) behind the spinal cord at the level between C4 and C7 (Figure 3A). This abnormal HDA was located in the posterior part on the right side in axial CT images (Figure 3B). MRI analysis was performed because the presence of a hemorrhagic lesion was suspected, based on the CT findings of the cervical bone. Sagittal T1-weighted MRI showed an iso-intensity mass in the epidural space behind the spinal cord at the level from C4 to C7 (Figure 4A). The lesion was noted as a heterogeneous high-intensity mass in sagittal T1-weighted images (Figure 4B). Axial T1-weighted MRI showed an iso-intensity mass in the posterior part on the right side in the epidural space and mild compression of the spinal cord (Figure 5A). The lesion was noted as a heterogeneous high-intensity mass in axial T2-weighted images (Figure 5B).

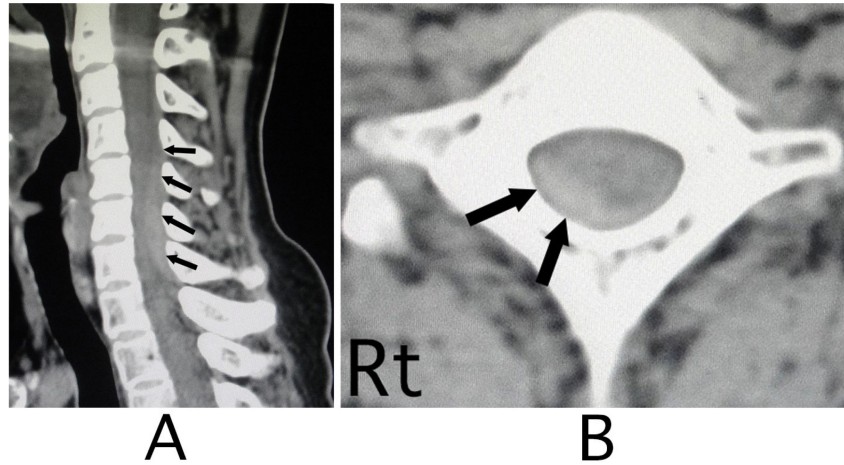

**Figure 3.** CT images of the cervical bone. CT images of the sagittal position showed a slight high-density mass behind the spinal cord at the level between C4 and C7 (black arrows) (**A**). The mass was located diagonally posterior part on the right side (black arrows) (**B**). (Rt = right).

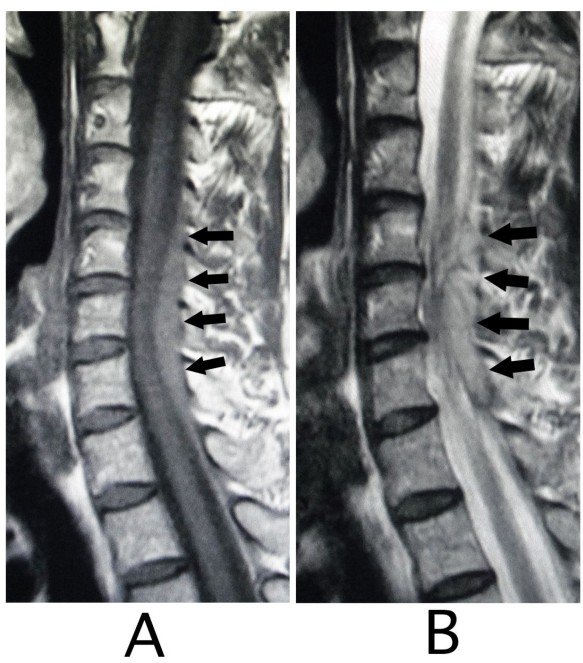

**Figure 4.** Sagittal MRI. An iso-intensity mass was identified in the epidural space behind the spinal cord at the level from C4 to C7 (black arrows) (T1-weighted image; (**A**)). The lesion was visualized as a high-intensity mass with heterogeneity (black arrows) (T2-weighted image; (**B**)).

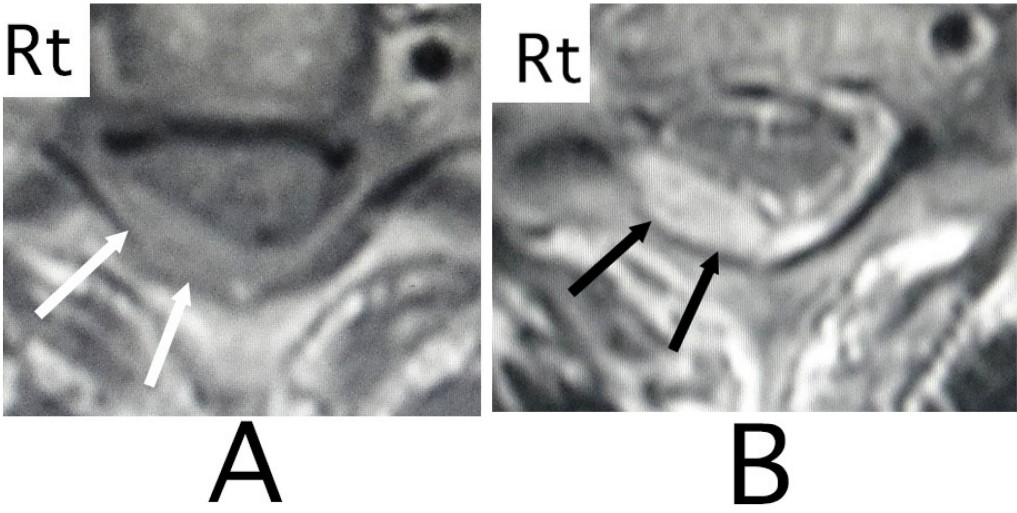

**Figure 5.** Axial MRI. An iso-intensity mass was identified in the diagonally posterior part on the right side (white arrows) (T1-weighted image; (**A**)). The lesion was visualized as a heterogeneous high- intensity mass (black arrows) (T2-weighted image; (**B**)). Mild compression of the spinal cord by the lesion was shown (**A**,**B**). (Rt = right).

She was diagnosed as having SSEH and hospitalized. After hospitalization, numbness and palsy improved by day 2 of hospitalization without operation. After discharge, the hematoma disappeared in the follow-up MRI analysis.

## 3. Discussion

The most common site of hemorrhage in SSEH is the posterior epidural space at the cervicothoracic level (C5 to Th2) and the thoracolumbar level (Th10 to L2) [2,5,6,8,10,13]. Hematoma of SSEH is delineated as an iso-intensity mass on T1-weighted MRI images and a hyper-intensity mass with inside heterogeneity (hypointense-foci) on T2-weighted MRI images in the acute stage within the first 24 h [1,3,4,8,13–15]. After 24 h, hematoma of

SSEH is delineated as a hyper-intensity mass on T1- and T2-weighted MRI images [5,12,14], and then the lesion was delineated as a low-intensity mass on T1- and T2-weighted MRI images [5,12,14]. MRI findings in our patients were consistent with those of SSEH in the acute stage within the first 24 h [1,3,4,8,13–15]. The diagnosis in our patients was SSEH, based on the MRI findings and the absence of traumatic or iatrogenic experiences in their clinical courses; however, the vertebrae levels of the lesions were inconsistent with the characteristics of SSEH [2,5,6,8,10,13].

The most frequent presentation at onset of SSEH is acute pain in the region from the back to the neck [1,2,4–6,8,10,12,13]. Back pain was present in both of our patients (Table 1), which was consistent with the characteristic onset of SSEH [1,2,4–6,8,10,12,13]. Rapidly progressive neurological deficits are caused by an accumulation of hematoma within the epidural space [1,2,4–6,8,15]. The location of hematoma shows a tendency to have an uneven distribution in the posterior part of the epidural space [6,9,12]. Accordingly, clinical features of SSEH depend on the site of hematoma [9,12] and include radiculopathy or myelopathy, or both, which are associated with various distributions of palsy of the extremities [2,6,15,16]. Patients without palsy of the extremities have also been reported [16]. The symptoms in each patient were consistent with acute myelopathy in patient 1 and acute radiculopathy in patient 2 in our case series. The cause of symptoms in each patient was compression of the spinal cord by hematoma in patient 1 and was considered to be compression of the lateral radicles, predominantly the posterior ones, by hematoma in patient 2. The symptoms in each patient correlated well with the location of their hematoma.

Predisposing factors of SSEH includes coagulopathy, administration of anticoagulant drugs or antiplatelet drugs, presence of minor trauma, coughing, blood dyscrasia, tumors of the spinal cord, hypertension, vascular malformation, and pregnancy [1–5,10,11]. In our patients, administration of aspirin prior to onset and the presence of hypertension were recorded in patient 1. There was a possibility that the administration was associated with an expansion of hematoma size in this patient.

The distribution of veins resembles that of arteries in the spinal cord [17]. Veins in the spinal cord are called the internal spinal cord vein [17]. The internal spinal cord vein is classified into the central vein and the peripheral vein (Figure 6) [17]. The central vein mainly collects venous blood in the gray matter and drains into the anterior medial vein. The distribution of the central vein is slightly smaller than that of the central artery [17]. The peripheral vein collects the venous blood in the white matter and approaches the surface of the spinal cord. After that, the peripheral vein drains into the coronary vein. There are anastomoses between the central vein and the peripheral vein. The anterior radicular vein originates from at the anterior part of the coronary vein and runs along the anterior radicle [17]. Similarly, the posterior radicular vein originates from the posterior part of the coronary vein and runs along the posterior radicle [17]. The internal vertebral venous plexus exists along the wall of the vertebral foramen in the epidural space around the entire circumference (Figure 6) [17]. The anterior radicular vein and the posterior radicular vein originate from the veins around the surface of the spinal cord (Figure 6) [17]. The anterior radicular vein and the posterior radicular vein converge and form the intervertebral vein (Figure 6) [17]. After the formation, the intervertebral vein connects with the internal vertebral venous plexus and drains into the segmental vein (Figure 6) [17]. The basivertebral vein collects venous blood inside the vertebral bone and connects with the internal vertebral venous plexus [17]. The internal vertebral venous plexus is located in the epidural cavity. The internal vertebral venous plexus connects with the external vertebral venous plexus and drains into the segmental vein [17]. The most acceptable origin for SSEH has been considered to be hemorrhage originating from rupture of the internal vertebral venous plexus (Figure 6) [1,4–6,8,10,12]. The internal vertebral venous plexus is well developed in the dorsolateral part [1,9,14,17]. The veins in the internal vertebral venous plexus are valveless with thin walls, additionally intravenous pressure of the internal vertebral venous plexus is low [1], which makes it vulnerable to sudden changes in pressure [8,9]. An increase in the intravenous pressure caused by Valsalva maneuvers may possibly cause the

internal vertebral venous plexus to rupture [1]. The posterior part of the internal vertebral venous plexus is considered to be the most likely point of rupture in SSEH [1,5–8,12,14]. The anterior part of the internal vertebral venous plexus is thought to be hard to rupture compared with the posterior part because the anterior part clings to the posterior longitudinal ligaments [1,4,14,15]. Hematomas in our patients were located posteriorly to the spinal cord, which was consistent with the characteristics of SSEH [1,5–8,12,14] and was probably caused by the rupture of the internal vertebral venous plexus.

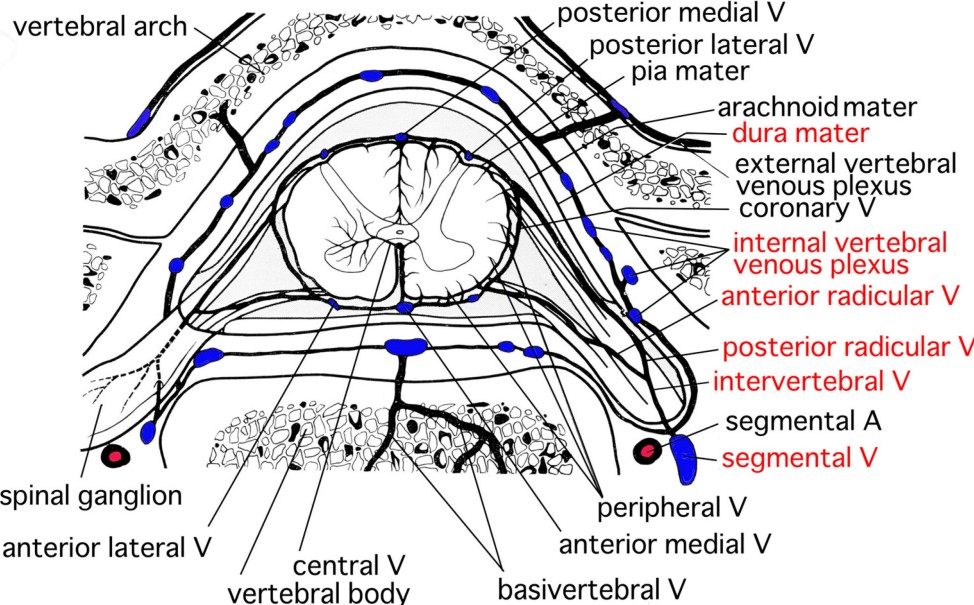

**Figure 6.** Venous perfusion of the spinal cord (modified from the original figure by Goto [17], we declare that the copyright of the original figure belongs to the late author Professor Goto). The anterior radicular vein and the posterior radicular vein converge and lead to the intervertebral vein. The intervertebral vein connects with the internal vertebral venous plexus in the epidural space and approaches the segmental vein. (A = artery, V = vein).

Once diagnosed with SSEH, urgent surgical intervention has been indicated in principle [1,2,5–8,18] and recommended within 48 h after onset [1,2,8]. SSEH with three factors, i.e., (1) severe neurological deficit, (2) significant mass effect, and (3) worsened clinical manifestation, is indicative for surgical treatment [4]. Decompressive laminectomy and hematoma evacuation are the standard surgical procedures [1,2,4,6–8,18]. Preoperative neurological deficits and interval between onset and operation were correlated with outcomes [3,4]. On the other hand, conservative treatment has recently been documented [1,2,4–6,12]. It was employed in patients with improving neurological deficits or minimal symptoms [10,13]. Improvement of neurological deficits without surgical treatment has been considered to associate with reduction of cellular edema by steroid therapy [6], the gradual spread of hematoma throughout the epidural space [8], and the leakage of liquid hematoma through the intervertebral foramen [16], which lead to decompression on the spinal cord [6,8,16]. Surgical treatment was not adopted in our patients because neurological symptoms got the improving course soon after hospitalization. Steroid therapy in the conservative therapy of SSEH has been reported [4,10]. Acceleration of recovery was speculated in response to steroid therapy [4]. There was a possibility that the administration of prednisolone accelerated the improvement of myelopathic symptoms in patient 1 in whom steroid therapy was performed.

## 4. Conclusions

Two patients who were diagnosed as having SSEH were shown. Their neurological abnormalities occurred acutely at onset, secondary to back pain. The clinical features

were associated with myelopathic signs in one patient and radicular signs in the other patient. The neurological findings in each patient correlated well with the location of the hematoma. Emergent surgical treatment was not adopted because progression of the neurological abnormalities was absent. Two patients carried good prognosis without surgical operations. In parallel, disappearance of the hematoma was also confirmed in the follow-up MRI analysis. SSEH is rare but should be considered in patients with myelopathy or radiculopathy with acute onset of symptoms. Our patients achieved a better clinical course after admission and showed good prognosis without being surgically operated on. MRI was useful in diagnosing SSEH in our patients. Additionally, the usefulness of multidirectional CT scans of the spinal cord was shown for the diagnosis of SSEH.

**Author Contributions:** K.O. summarized the neurological and radiological findings for each patient and designed the context of the article. K.O. and T.A. identified the contents of medical charts and CT/MRI findings and developed the discussion part. K.O. and M.H. evaluated the CT/MRI findings and correlations between locations and neurological symptoms in each patient. M.F. provided the advice on the discussion part. H.N. checked the contents of the article and corrected the summary and discussion. H.U. evaluated the neurological findings and MRI findings, and judged the necessity of the surgical operation. All authors have read and agreed to the published version of the manuscript.

**Funding:** This research received no external funding.

**Institutional Review Board Statement:** This study was approved by the Institutional Research Review Board of Nihon University Hospital (20230205).

**Informed Consent Statement:** Informed consent was obtained from each patient to publish this work.

**Data Availability Statement:** Not applicable.

**Acknowledgments:** We thank all members of the Division of Neurology, Department of Medicine, Nihon University School of Medicine, for their advice and help with the study. We also thank Masaru Kushimoto who is the chief of the Emergency Room (ER) in Akabane Central General Hospital.

**Conflicts of Interest:** The authors declare no conflict of interest.

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
