# Peer review of "Two Patients with Spontaneous Spinal Epidural Hematoma Carrying a Good Prognosis without Surgical Operations"

_2035-8377, doi:10.3390/neurolint15010024_

Round 1
Reviewer 1 Report
SSEH is a rare neurological emergency that requires surgical treatment in most cases, but conservative treatment has been reported previously.
This paper presents two SSEH cases who improved after conservative treatment.
Compared with previously published articles, the disease characteristics of the two patients described in this article do not conform to the typical clinical manifestations of SSEH, which is of great significance for clinical work and is conducive to a more comprehensive understanding of SSEH.
This paper has strong originality and standard writing. The medical records provided in this article are novel and detailed, and the pictures and tables are concise, clear, detailed and accurate. Iare instructive for clinical work.
The content of the article is novel and detailed, the writing is standardized, and the citation is appropriate.
It would be better if there was spinal angiography, which could determine whether it was related to the internal vertebral venous plexus, and whether there was hemangioma, vascular malformation, etc
Author Response
Thank you for considering our paper entitled “Two Patients with Spontaneous Spinal Epidural Hematoma Carrying a Good Prognosis without Surgical Operations”, by Katsuhiko Ogawa, Takayoshi Akimoto, Makoto Hara, Midori Fujishiro, Hiroshi Uei, andHideto Nakajimafor publication Case Report in your journal, subject to the first revision.
Enclosed are one original of the revised manuscript. The reviewers’ comments helped us to improve our paper, which we have revised accordingly by attempting to address the questions posed and by taking the advice given, in the following way:
Yours sincerely,
Katsuhiko Ogawa, MD
In Response to Reviewer 1’s Comments:
Spinal artery angiography
Unfortunately, angiography of spinal cord artery was not preformed in either patient.
Reviewer 2 Report
With regard to the article entitled” Two Patients with Spontaneous Spinal Epidural Hematoma Carrying a Good Prognosis without Surgical Operations”
1.There was fine grammatical sufficiency and reasonable logism in the abstract with regard to the SSEH.
2.In the introduction, authors described the clues of exclusion of secondary SSEH. They wrote the workflow and feature of this rare but devastating disease entity.
“conscents”.
4. In the Presentation, two patient were well addressed in their symptom, neurological deficit, diagnostic thinking, hospital course and eventual recovery.
5. In the Conlcusion, the report shed a light about the SSEH with subsequent improved neurological presentation might tolerate the time-sensitive surgical intervention. In clinical practice guideline, once diagnosed with SSEH, urgent surgical intervention is usually indicated within 48 hours after onset
Preoperative neurological deficits and interval between onset and operation were correlated with outcomes. However conservative treatment has recently been documented in some series, it was employed in patients with improving neurological deficits or trivial symptoms. Some myth about the unresolved explaining about the “steroid therapy in the conservative therapy of SSEH”, and “the role of modulating mechanism” deserves further study interest in the further time. Acceleration of recovery was doubted in responsive to steroid therapy and may authors consider more possible mechanism about the medical cord decompression without hematoma evacuation.
6.In summary, I agree with the authors: the limited findings from the current literature body support possible viewpoints that some possibility that the conservative medical treatment may assist the improvement of myelopathy. Since the surgical indication is decided by case-to-case basis for this acute SSEH, we still need to clarify the pathophysiology of this rare disease.
Author Response
Thank you for considering our paper entitled “Two Patients with Spontaneous Spinal Epidural Hematoma Carrying a Good Prognosis without Surgical Operations”, by Katsuhiko Ogawa, Takayoshi Akimoto, Makoto Hara, Midori Fujishiro, Hiroshi Uei, andHideto Nakajima for publication Case Report in your journal, subject to the first revision.
Enclosed are one original of the revised manuscript. The reviewers’ comments helped us to improve our paper, which we have revised accordingly by attempting to address the questions posed and by taking the advice given, in the following way:
Yours sincerely,
Katsuhiko Ogawa, MD
In Response to Reviewer 2’s Comments:
The possible mechanism of decompression on the spinal cord
- We added “Improvement of neurological deficits without surgical treatment has been considered to associate with reduction of cellular edema by steroid therapy [6], the gradual spread of hematoma throughout the epidural space [8], and the leakage of liquid hematoma through the intervertebral foramen [16], which lead to decompression on the spinal cord [6, 8, 16].”(Page 6, line 12 to 15)
Reviewer 3 Report
In this article, the authors present two cases of SSEH that were successfully managed without requiring any surgical procedure. They describe both cases in detail; they present an extensive literature review on the subject. For me, there is only one point that would require further clarification. In page 1 line 41, they write: "Emergent surgical treatment is indicated in patients with progressive neurological deficits. Evacuation of the hematoma for decompression of the spinal cord remains a standard surgical procedure". The management of patient 1 is therefore somewhat surprising. The patient had a severe neurological deficit and an epidural mass compressing the spinal cord was found on MRI. The neurological deficit started improving (only partially) on day 4. Can the authors explain better why it was not classified as an emergency and a surgical decompression was not considered? What were the characteristics of the case that led you to expect a satisfactory outcome?
Author Response
Thank you for considering our paper entitled “Two Patients with Spontaneous Spinal Epidural Hematoma Carrying a Good Prognosis without Surgical Operations”, by Katsuhiko Ogawa, Takayoshi Akimoto, Makoto Hara, Midori Fujishiro, Hiroshi Uei, andHideto Nakajima for publication Case Report in your journal, subject to the first revision.
Enclosed are one original of the revised manuscript. The reviewers’ comments helped us to improve our paper, which we have revised accordingly by attempting to address the questions posed and by taking the advice given, in the following way:
Yours sincerely,
Katsuhiko Ogawa, MD
In Response to Reviewer 3’s Comments:
No adoption of the surgical treatment
- We changed “On day 4 of hospitalization, paraplegia, numbness, and decreased micturition disappeared,” to “After hospitalization, paraplegia, numbness, and decreased micturition gradually improved, and then, these neurological symptoms disappeared on day 4 of hospitalization,” (Page 4, line 6 to 8)
- We changed “neurological symptoms improved after onset.” to “neurological symptoms got the improving course soon after hospitalization.”. (Page 6, line 15 to 16)